# Oral Rehabilitation as Part of a Multidisciplinary Treatment in a Case Study of Pigmentary Incontinence

**DOI:** 10.3390/children10091505

**Published:** 2023-09-04

**Authors:** Mónica Cano-Rosás, Joaquín de Vicente-Jiménez, José María Diosdado-Cano, David Suárez-Quintanilla, Rogelio González-Sarmiento, Daniel Curto, Adrián Curto

**Affiliations:** 1Department of Surgery, Faculty of Medicine, University of Salamanca, Alfonso X El Sabio Avenue s/n, 37007 Salamanca, Spain; mcanorosas@usal.es (M.C.-R.); jvj@usal.es (J.d.V.-J.); 2Faculty of Dentistry, University of Sevilla, Avicena Street s/n, 41009 Sevilla, Spain; jmdcodontology@outlook.es; 3Department of Surgery, Faculty of Medicine and Dentistry, University of Santiago de Compostela, San Francisco Street s/n, 15782 Santiago de Compostela, Spain; david.suarez@usc.es; 4Department of Medicine, Faculty of Medicine, University of Salamanca, Alfonso X El Sabio Avenue s/n, 37007 Salamanca, Spain; gonzalez@usal.es; 5Department of Patholoy, 12 de Octubre University Hospital, Córdoba Avenue s/n, 28041 Madrid, Spain; daniel.curto@salud.madrid.org

**Keywords:** Bloch–Sulzberger syndrome, pigmentary incontinence, facial hemiatrophy, orthodontics

## Abstract

We present the clinical course of a 9-year-old female patient with Bloch–Sulzberger syndrome and severe neurological deficit that met the major (classic cutaneous signs) and minor (dental anomalies and retinal pathology) diagnostic criteria of Landy and Donnai. Longitudinal multidisciplinary follow-up was carried out from birth to adulthood. Neurological involvement was assessed with electroencephalographic (EEG) and neuroimaging tests at different times during the patient’s life. Cranio-maxillofacial involvement was evaluated using lateral skeletal facial and cephalometric analyses. The right and left facial widths were measured through frontal face analysis and using the vertical zygomatic–midline distance. Oral rehabilitation was performed through orthodontic treatment and major dental reconstruction using composite resins. This treatment aimed to improve the occlusion and masticatory function, relieve the transversal compression of the maxilla, and reconstruct the fractured teeth. We believe that, due to significant neurological and cognitive impairment, orthognathic surgery was not the best option for restoring function and improving oral health-related quality of life.

## 1. Introduction

In 1926, a clinical case resembling pigmentary incontinence (PI) was reported [1]. Subsequently, Bloch (1926) and Sulzberger (1928) described the eponymous syndrome. PI, also known as incontinentia pigmenti or Bloch–Sulzberger syndrome, is a rare genetic disease with a dominant transmission pattern linked to the X chromosome caused by a molecular alteration in the NEMO gene located at the Xq28 locus [2].

The most common cause of PI is a recurrent genomic reorganization in NEMO, which causes the defective activation of transcriptional nuclear factor kappa beta (NFkB). NFkB is fundamental in many immune, inflammatory, and apoptotic pathways. The most frequently noted mutation is a deletion of NEMO exons 4–10 [3].

The birth prevalence rate of PI is estimated to be 1.2/40,000 [4]. PI almost exclusively affects women, and it is usually lethal in the prenatal development of males, as they cannot survive without a normal copy of the NEMO gene [5]. However, some cases of males who survived because they presented somatic mosaicism or trisomy XXY have been reported in the literature [6,7,8]. Although this pathology is X-linked, it is usually a denovo mutation, and PI affects all ectodermal cell lines, i.e., skin, nails, hair, and teeth [4].

The syndrome is characterized by four classic cutaneous stages: vesicular or inflammatory phase 1, verrucous phase 2, hyperpigmentary phase 3, and hypopigmented or cicatricial atrophic phase 4. These classic stages evolve chronologically. However, some patients may not present all phases, or some may overlap temporarily as they appear. The skin lesions follow the pattern of Blaschko’s lines [9].

## 2. Case Report

A 9-year-old girl with Bloch–Sulzberger syndrome was referred to the stomatological unit of the University Clinical Hospital of Valladolid, where she had a multidisciplinary follow-up in collaboration with professionals from the University of Salamanca, Spain.

The patient, with no family history of dermatological conditions, suffered a neonatal episode of clonic seizures and a skin rash with a linear distribution along the extremities. The gestation was considered high risk, as the pregnancy resulted from in vitro fertilization. The pondostatural development at birth was normal, with a slightly above-average cranial perimeter (36 cm). During the following two weeks, the exanthema evolved into linear vesicular lesions, mainly on the lower limbs. At one month of life, the lesions became keratinized, giving rise to the typical hyperkeratotic lesions of the verrucous phase (Figure 1A).

A neonatal cytogenetic study was performed on peripheral blood lymphocytes using the GTG banding technique (550–850 bands), which showed an apparently normal number and structure of chromosomes, with a female karyotype of 46 XX.

The neuropediatric results of a brain computerized axial tomography (CAT) carried out at birth revealed left temporal–parietal cortical dysgenesis, and an EEG revealed interhemispheric asymmetry. Adequate pondostatural growth was observed in the growth curves from 0 to 2 years of age. However, the growth of the cranial perimeter was below normal values, indirectly indicating a lack of brain development.

Ophthalmologists diagnosed retinopathy in both eyes at birth, which evolved favorably. At 5 years of age, the retinopathy remained stable, but divergent strabismus and microphthalmia of the right eye were diagnosed.

At 8 years of age, the patient presented linear hyperpigmented areas in the axilla and trunk (Figure 1B).

There was a delay in psychomotor development that affected both cognition and language (anarthria requiring speech therapy), as well as motor skills (spastic paraparesis in the lower extremities that caused thinning of the right leg with a smaller foot). The patient did not attend planned check-ups with a speech therapy specialist and did not follow the specialist’s instructions. At 10 years of age, less development of the right leg was evident, and there were faded bruises resulting from frequent falls suffered by the patient (Figure 2A).

At 15 years of age, the hypotrophy of the right leg was very apparent in the standing position and from behind (Figure 2B). This difference is presented in Figure 2C, a drawing of Figure 2B, which shows the perimeter of both legs measured using a measuring tape. Figure 2D, as well as a drawing of Figure 2B, illustrates the hypochromic linear spots. In this image, the left leg is superimposed onto the right to visualize the size difference caused by the hypotrophy of the right leg.

A lack of motor coordination caused instability when the child walked, which led to a traumatic fall at the age of 8 that fractured both upper central incisors. There were alterations in the dental morphology involving the conoid lateral superior left incisor and the presence of supplementary cusps in both upper lateral incisors (Figure 3).

At the age of 9, oral rehabilitation was initiated with neurological tests and a complete dental, facial, and cephalometric analysis. The seizures were treated with sodium valproate and had not reoccurred since the neonatal period. Also, at 9 years of age, an EEG showed the existence of points (*) in the left hemisphere with phase inversion in the left central region (enlarged in the lower right part of the figure) that did not exist in the corresponding channels of the right hemisphere (Figure 4).

At 10 years of age, neuroimaging tests with magnetic resonance were performed. In an axial section, a large poroencephalic cyst was observed that occupied nearly the entire left frontal lobe, and a much smaller cyst in the right frontal lobe adjacent to the Sylvian fissure. There was periventricular white matter hyperintensity and diffuse cranial hyperostosis (Figure 5). In a coronal section, cranial asymmetry and cerebral parenchyma atrophy were observed, with marked grooves in the temporal and parietal lobes and an increase in the size of the cerebral ventricles (Figure 5).

Although the frequency with which PI patients present agenesis, hypodontia, and even anodontia, and skeletal class III with loss of vertical dimension has been reported in the literature [10,11], our patient had no missing teeth and presented a skeletal class II with an increase in the vertical dimension (ENA-Xi-Pm > 47°) and convex profile [10,11,12,13].

The frontal facial analysis showed disharmony among the facial thirds (Figure 6A); vertical facial asymmetry was manifested as a lack of parallelism between the interciliary, interpupillary, infraorbital, and intercommissural lines (Figure 6B). Transverse facial asymmetry was manifested as the distance between the inner corner of the eye and the corner of the mouth on the right side being shorter than on the left side (Figure 6C). The chin deviated from the mid-sagittal plane toward the left side (side shorter and of lower condylar growth) both in maximum intercuspation and centric relation and in maximum opening (Figure 6C). The gonial angle was higher on the left side, with skeletal mandibular asymmetry (Figure 6D). In the facial profile analysis, labial protrusion was observed with respect to Ricketts’ esthetic plane and the convex profile (Figure 6E). In addition, right facial hemiatrophy (the distance between the zygoma and the vertical midline was less on the right side than on the left) and an ogival palate were observed. A cephalometric analysis was performed according to Steiner and revealed a skeletal class II with mandibular retrognathism and posterorotation (Figure 6F). The patient’s lateral radiographs were always taken in the same radiological center obtained using the conventional method of positioning the head using a cephalostat head positioner and the Frankfurt horizontal plane parallel to the ground [14].

Due to significant neurological damage, orthognathic surgery was ruled out. It was decided to camouflage the skeletal discrepancy through two treatment phases. In the first phase, fixed Mini-Twin multibrackets with 0′22 Roth prescription slots were used for 7 months (Figure 7A). In the second phase, an expansion plate was placed for 8 months to increase the transversal dimension of the maxilla, improve the space available for the tongue, and stop mandibular posterorotation (Figure 7B). At 11 years of age, the orthodontic treatment was completed, and the repair of incisive fractures was performed using composite resins made of macro-filler because of its greater resistance to occlusal loads.

The clinical and radiographic follow-up showed no signs of recurrence, and at 18 years of age, the improved occlusion achieved with the orthodontic treatment had been maintained.

Although, over time, there was some alteration in the color of the restoration, both the parents and the patient were satisfied with the functional and aesthetic result. If the images at the onset of the treatment (9 years of age) and the end of treatment (18 years of age) are compared, it can be observed that improving the occlusion and suppressing the occlusal trauma also improved the gingival recession of the lower incisors (Figure 8). 

In terms of speech, it should be noted, as has been done previously, that the patient was told that she needed to see a speech therapist to correct this aspect, but the patient did not attend her follow-up visits with this specialist, and therefore, we cannot describe any improvement in this aspect. In terms of eyesight, the patient continued to have regular check-ups with an ophthalmology specialist. Regarding the strabismus, the ophthalmologist treating the patient considered it necessary to wait more years for a more invasive surgical treatment, considering that at the age of 18, it was still too early to perform such a treatment. The patient continued her regular check-ups with a dermatology specialist to monitor the evolution of her dermatological lesions. On a neurological level, the patient attended regular neurological rehabilitation treatment.

## 3. Discussion

Diagnostic criteria for PI were introduced by Landy and Donnai in 1993, who identified cutaneous signs, such as erythema, vesicular-bullous lesions, hyperpigmented striae, and linear alopecia striae, as being the most important criteria for diagnosing the syndrome. By contrast, dental, hair, and retinal abnormalities were considered less significant diagnostic criteria [15].

Neurological manifestations appear in 10–30% of cases and mark the prognosis of the disease, which can include epilepsy and a delay in psychomotor development. Brain atrophy, neonatal stroke, and corpus callosum hypoplasia have also been described [16,17].

The most frequent ocular alterations are retinal dysplasia; however, uveitis, atrophy of the optic nerve, strabismus, cataracts, and congenital retrolental fibroplasia have also been described [18].

Cranio-facial anomalies include cranial deformities, facial asymmetries, and transverse deficiency of the maxilla, micrognathia, and microphthalmia. The facial asymmetry is usually caused by hemifacial atrophy [9].

Oral manifestations appear in 90% of PI patients and may include taurodontia, agenesis, hyposialia, hypodontia, delayed eruption, and anomalies in tooth shape (conical teeth, pin teeth, accessory cusps) [16,19]. The ogival palate, cleft palate, cleft lip, swallowing, and voice disorders have also been described [12,16,20,21].

The dentofacial deformities associated with pigmentary incontinence have been described in the literature. However, dental examination is important since Bloch–Sulzberger syndrome is a congenital etiological factor of malocclusion [21].

The skeletal problems that have been reported include maxillary transversal deficit associated with oligodontia in both jaws [9,22], facial asymmetry [21], and skeletal class III malocclusion [10] with loss of vertical dimension and mandibular anterorotation [9,21].

Cases with class II malocclusion have also been described, but the results of these studies were not statistically significant because of the small sample size [12].

The facial asymmetry observed in this case is the same as that described in patients with PI by other authors [9,12].

The alteration of tooth shape and the existence of the ogival palate are also consistent with that described by other authors. In Sweden, 30 individuals, 29 females and 1 male, from 17 families with a clinical diagnosis of PI were studied, and alterations of tooth shape were found in 100% of the individuals examined [19]. In Serbia, 25 subjects belonging to nine families were studied, and 12% were found to have an ogival palate [16]. Several authors suggest that the high and deep palate may be a secondary feature of PI that is detectable during the first year of the patient’s life, while other dental alterations appear only after the first 12 months [16,20].

A significant number of articles refer to the frequent finding of a decrease in the number of teeth [16,19,21,22], although, in our patient, this was not the case.

Although the classic presentation of the syndrome is dermatological, seizures may arise, although infrequent, after birth during the perinatal period [17,23], as in the case presented here.

At the dermatological level, when there are area skin lesions, it is necessary to make a differential diagnosis with other dermatological pathologies, such as vesicular-bullous eruptions that are observed in herpes simplex infection or in other skin lesions that occur in impetigo or in other autoimmune pathologies (During’s herpetiform dermatitis or bullous pemphigoid) [4,9,24]. Nail dysplasia is observed in 40% to 60% of patients with this pathology [4].

Ignorance of this pathology means that many mild cases go unnoticed and reach adulthood without diagnosis [25].

One of the limitations of this clinical case is not being able to describe the entire clinical history of the patient from the primary dentition to the permanent dentition or not being able to describe the follow-up with the speech therapy specialist to assess her speech improvement.

## 4. Conclusions

The pattern of skeletal malocclusion in patients with PI can vary between class I, class II, and class III, the latter being more common in the presence of agenesis. Facial asymmetry is also frequent and has been described in the literature.

Skin changes tend to diminish in intensity over time and are sometimes not very evident. In these cases, the buccal–dental alteration becomes especially significant in the diagnosis since it remains throughout the patient’s lifetime. As dental anomalies are the second most frequent finding in PI patients, the role of dentists is vital, especially in those cases in which there are no skin alterations or skin alterations that are not evident.

Regarding the International Classification of Functioning and Disability (ICF), masticatory and functional alterations and the psychological and communicative barriers encountered by people with rare diseases can limit their ability to perform certain activities and restrict their participation in the community. Malocclusion and cranio-maxillofacial alterations can cause organic dysfunction, which can also have a psychological impact. On the other hand, skeletal problems and dentofacial deformities left untreated can contribute to a lower oral health-related quality of life, making it necessary for both maxillofacial surgeons and odonto-stomatologists to be integrated into the multidisciplinary team to carry out evaluations and early treatment.

Correcting malocclusions and possible oral–dental alterations can result in functional (the ability to chew and communicate) and aesthetic (integration and socialization with the environment) improvements.

## Figures and Tables

**Figure 1 children-10-01505-f001:**
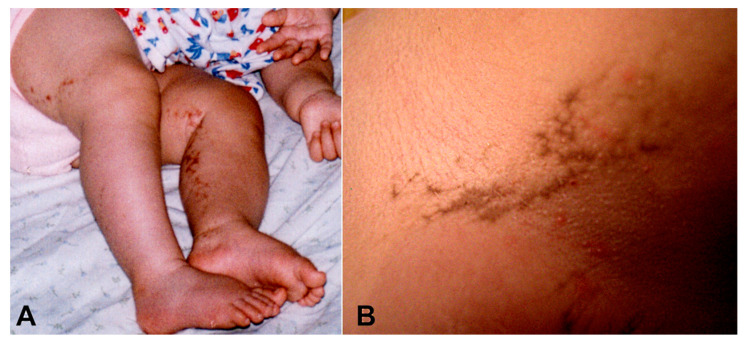
(**A**) Lesions in the verrucous phase on the lower extremities, 8 months of age. (**B**) Hyperpigmented linear areas in the axilla, 8 years of age.

**Figure 2 children-10-01505-f002:**
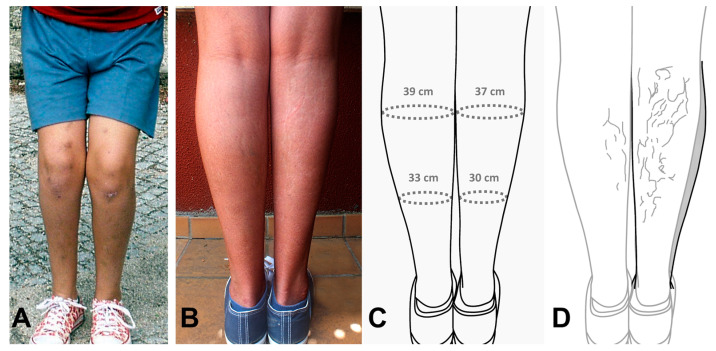
(**A**) Patient with less development of the right leg and bruising, 10 years of age. (**B**) Patient in the standing position and from behind. (**C**) Drawing of section B with representation of the perimeter of both legs. (**D**) Drawing of section B with a schematic representation of linear hypochromic spots, left leg superimposed (grey) to show the hypotrophy of the right leg.

**Figure 3 children-10-01505-f003:**
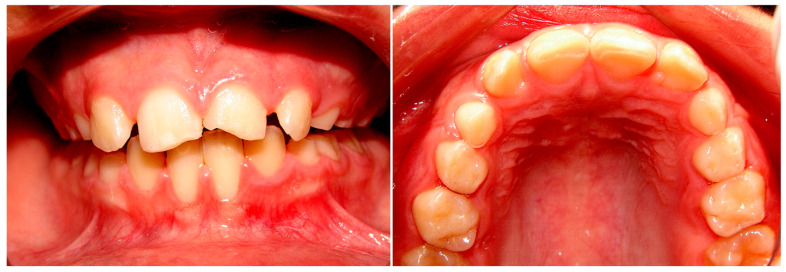
Fracture of the central incisors, accessory cusps in upper lateral incisors, and upper lateral incisor conoid. Upper occlusal view taken using a mirror.

**Figure 4 children-10-01505-f004:**
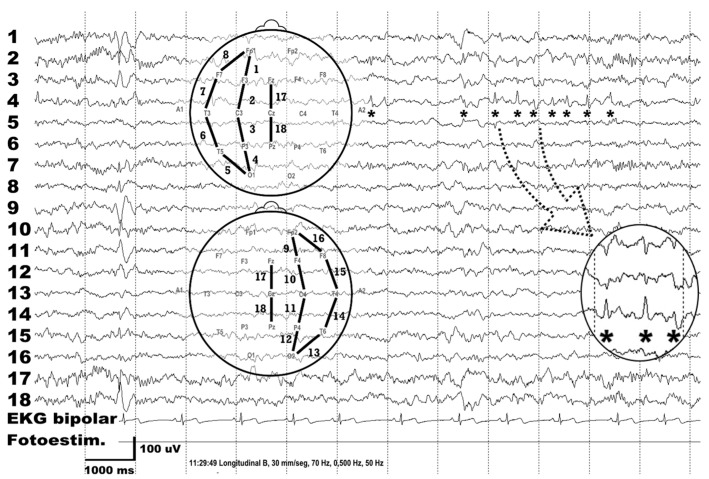
EEG tracing at 9 years of age showing points (*) in the left hemisphere with phase inversion in the left central region (enlarged in the lower right part of the figure). EKG: electrocardiogram; fotoestim: photostimulation; uV: microvolts; ms: milliseconds. Conditions of the EEG register: recording speed of 30 mm per second, filters of high and low frequencies at 70 and 0.5 hertz, respectively, and network filter of 50 Hz activated.

**Figure 5 children-10-01505-f005:**
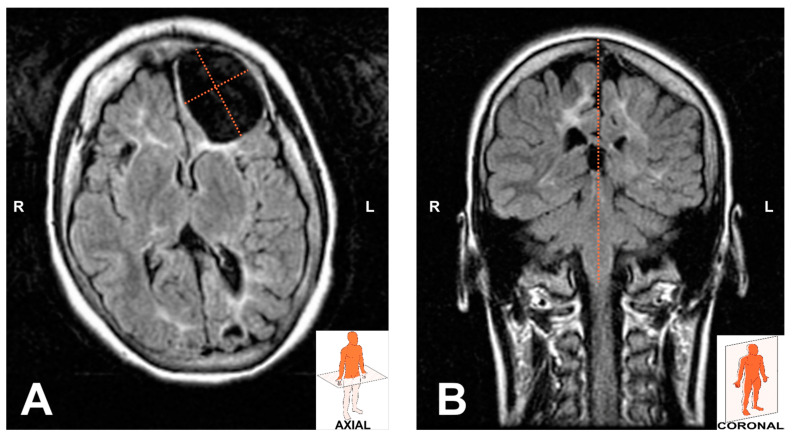
NMR at 10 years of age. (**A**) Axial section (poroencephalic cysts, hyperintensity of periventricular white matter, and diffuse cranial hyperostosis). (**B**) Coronal section (cranial asymmetry and parenchymal atrophy). NMR: nuclear magnetic resonance.

**Figure 6 children-10-01505-f006:**
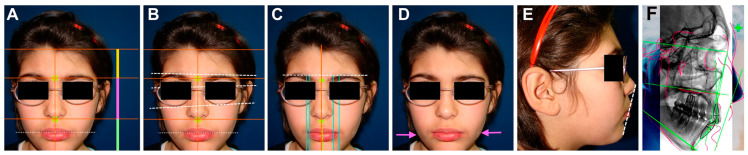
Frontal facial analysis with vertical facial asymmetry (**A**,**B**) and transverse (**C**) and higher left goniaco angle (**D**); facial profile analysis with labial protrusion (**E**); skeletal class II malocclusion with retrognathism and mandibular posterorotation identified with Steiner cephalometric analysis (**F**).

**Figure 7 children-10-01505-f007:**
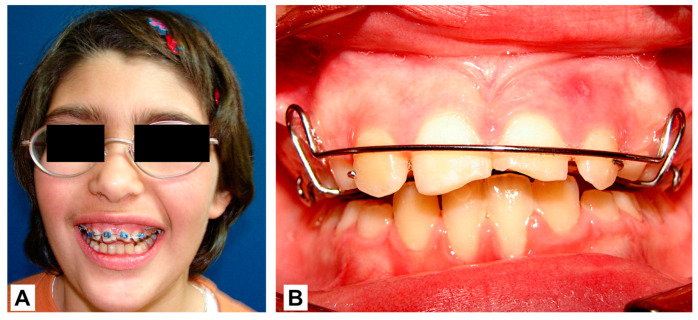
Orthodontic treatment in two phases: (**A**) Fixed multibrackets for 7 months. (**B**) Expansion plate for 8 months.

**Figure 8 children-10-01505-f008:**
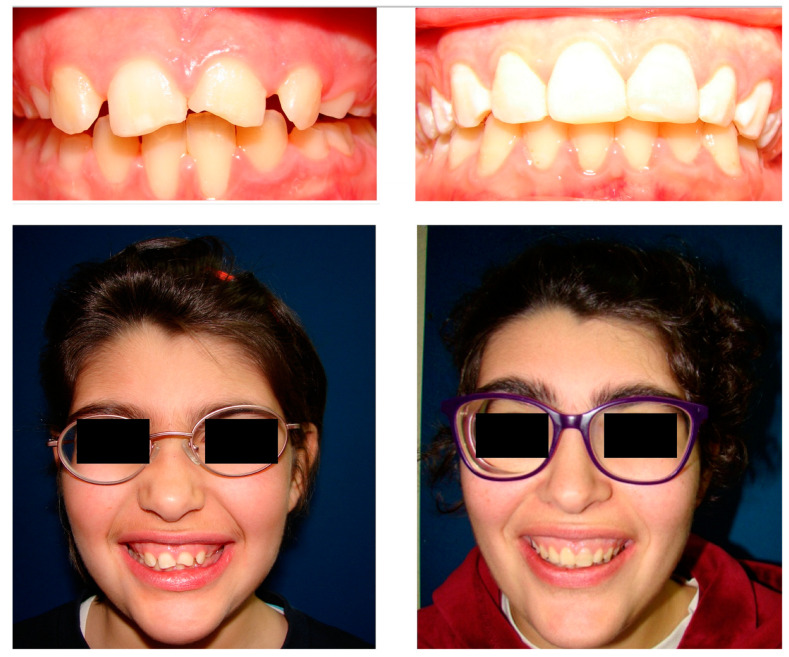
Start (9 years) and end (18 years) of the dental treatment.

## Data Availability

No data in the paper reveal the patient’s identity. The data presented in this study are available upon request from the corresponding author.

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
