# Peer review of "Oral Rehabilitation as Part of a Multidisciplinary Treatment in a Case Study of Pigmentary Incontinence"

_children, 2023, doi:10.3390/children10091505_

Round 1

Reviewer 1 Report

 „European birth prevalence rate of PI is estimated to be 1.2/100,000 [4].“ Pls check world wide and state other estimations ( e.g. https://eyewiki.aao.org/Incontinentia_Pigmenti)

Pls extend the discussion.

minimal changes required

Reviewer 2 Report

Parent advocacy groups and medical information tend to favor using Incontinentia pigmenti.  
Title should be reconsidered.

There are referenes for worldwide prevalance of IP instead of just focusing on European prevalance.

It would be helpful to add that the while the syndrome/disorder is x-linked it is usually a denovo mutation and
PI affects all ectodermal cell lines skin, nails, hair, and teeth.  This would capture the multidiscplinary care need
in the Introduction.

In the Discussion the four classic cutaneous stages are brought up but the article does not bring up the specifics
of the patient in the discussion.  This paragraph in the discussion may be better served in the
intro since the authors refer to the verrocous phase in Fig 1.  This should be defined for the reader more clearly.

[Important] The case report brings up a 9 year reporting to the academic hospital but discussions are made prior to this
time and after.  It is important to document the source of the information (chart review,dicussions with treating physicians).
Some aspects are incomplete, for example a lack of dental records in the primary tooth dentition and follow up of full permanent teeth
dentition.  There isn't a discussion if later posterior teeth had delayed eruption and comments about a lack of micro/hypdontia should be documented.  

[Important] There is a lack of followup from the anarthria at age 8 and changes to speech at later ages.  For example, anarthia at
this age is quite severe and what was the speech intervention and what progress was made.

[Important] A more organized approach is needed to document source information and cite why there are deficiencies in some follow up.

[Important]  the final case report section should end with a full analysis at 18 instead of ending with the 9 year orthodontic treatment.
The full analysis should include speech, eye, skin, and neurological.

Discussion
In the case results it is mentioned that the child had parents that did not have any skin lesions.
There is an opportunity in the Discussion to discuss how the x-link disorder occurs mainly from de novo mutation
and that the case is typical to other cases in this regard.  

Reviewer 3 Report

The paper Oral Rehabilitation as Part of Multidisciplinary Treatment in a Case Study Involving Bloch–Sulzberger Syndrome is well written, well documented, and high-quality images are presented. 

However, there are some issues that need further attention: 

The caption in Fig. 6, is Steiner, not Steinert. 

The orthodontic treatment included in the first phase fixed appliances, followed by an expansion plate. Usually, it is done the other way around. Could you please explain why you have chosen this type of treatment? 

Composite resin causes pulp necrosis, why have you chosen composite at the age of 11?

Orthognathic surgery was ruled out, however, in the abstract authors point out that orthognathic surgery was the best option for restoring function and improving oral health-related quality of life. Could you explain?
